# Volatiles Composition and Antimicrobial Activities of Areca Nut Extracts Obtained by Simultaneous Distillation–Extraction and Headspace Solid-Phase Microextraction

**DOI:** 10.3390/molecules26247422

**Published:** 2021-12-07

**Authors:** Martina Machová, Tomáš Bajer, David Šilha, Karel Ventura, Petra Bajerová

**Affiliations:** 1Department of Analytical Chemistry, Faculty of Chemical Technology, University of Pardubice, Studentská 573, 532 10 Pardubice, Czech Republic; martina.machova@student.upce.cz (M.M.); karel.ventura@upce.cz (K.V.); petra.bajerova@upce.cz (P.B.); 2Department of Biological and Biochemical Sciences, Faculty of Chemical Technology, University of Pardubice, Studentská 573, 532 10 Pardubice, Czech Republic; david.silha@upce.cz

**Keywords:** areca nut volatiles, HS-SPME, SHDE, antimicrobial activity

## Abstract

The volatile components of areca nuts were isolated by headspace solid-phase microextraction (HS-SPME, DVB/CAR/PDMS fiber extraction) and simultaneous hydrodistillation–extraction (SHDE) and analyzed by gas chromatography/mass spectrometry. Furthermore, all SHDE fractions were tested for antimicrobial activity using the disk diffusion method on nine Gram-negative and Gram-positive bacteria (*Bacillus subtilis*, *Enterococcus faecalis*, *Escherichia coli*, *Pseudomonas aeruginosa*, *Staphylococcus aureus*, *Streptococcus agalactiae*, *Streptococcus canis*, *Streptococcus pyogenes*, and *Candida albicans*). In total, 98 compounds (mainly alcohols, carbonyl compounds, fatty acids, esters, terpenes, terpenoids, and aliphatic hydrocarbons) were identified in SHDE fractions and by using SPME extraction Fatty acids were the main group of volatile constituents detected in all types of extracts. The microorganism most sensitive to the extract of the areca nut was *Streptococcus canis*. The results can provide essential information for the application of different treatments of areca nuts in the canning industry or as natural antibiotics.

## 1. Introduction

*Areca catechu* is a medium-sized palm tree and is part of the Arecaceae family, which contains more than 200 genera and about 2600 species. This palm tree is commonly known as areca palm, areca nut palm, betel palm, betel nut palm, Indian nut, Pinang palm, or catechu. The betel nut (or areca nut) is a seed of the fruit of the palm tree *Areca catechu*. This palm tree is widely distributed in South and Southeast Asia, including China, India, Indonesia, Malaysia, the Philippines, and New Guinea. The traditional habit in these countries is to chew betel quid, which contains betel leaf, areca nut, and slaked lime (calcium hydroxide) and can contain tobacco. Furthermore, other substances, particularly spices, such as cardamom, saffron, cloves, aniseed, turmeric, mustard, or sweeteners, are added according to local preferences [1,2,3]. Up to 10–20% of the world’s population (up to 600 million people) chews betel quid, making it the fourth most used drug, after alcohol, nicotine, and caffeine. Humans use this drug primarily for stimulating effects that are comparable to the use of nicotine [4,5]. Arecoline can also produce addiction and withdrawal symptoms, such as nicotine. The chewing of areca nuts causes many effects on the autonomic nervous system, such as euphoria, increased alertness, focused attention, sweating, stimulation of excitement, elation, and anxiolysis [6].

Chewing betel quid has been found to improve oral hygiene. People chewing betel quid have less dental caries compared to the nonusing population [7,8]. The substances contained in the areca nuts inhibit Gram-positive microorganisms that cause dental caries. The antimicrobial activity of different extracts of the areca nut was proved [9,10]. However, it should be mentioned that the chewing of the areca nut has negative effects on human health (high blood pressure, liver cirrhosis, and cardiovascular diseases) [11]. There are many studies [12,13,14] that have proven the carcinogenity of areca nuts.

Areca nuts contain many different groups of compounds such as alkaloids; the most known alkaloids are arecoline, arecaidin, guavacoline, and guavacin [15], and the newly identified alkaloids are homoarecoline and arecatemines [16]. Other compounds belong to the group of flavonoids [17], carbohydrates, lipids, proteins, crude fibers, and minerals [18]. Differences in the composition and concentration of substances in areca nuts can be caused by the different geographical location or degree of ripeness of the nut.

Despite the fact that areca nut is a widely known drug and plant, there are not enough reports dealing with the volatile profiles of this matrix. Only three studies analyzed compounds in areca nuts using GC-MS after different types of extraction techniques. One of them was Soxhlet extraction of green immature fruits [19]. The other was solvent extraction with dichloromethane after 24 h of soaking a small piece of betel nut in alkaline distilled water [20]. The last was the use of 50% aqueous acetone for the extraction of betel nuts, and the extract was revealed to be a mixture of fatty acids (lauric, myristic, palmitic, and oleic) [21]. By using high-performance liquid chromatography with different types of detectors, there were determined following groups of compounds in areca nuts or chewable products—alkaloids [15,18,22,23], polyphenols [18], and phenols [23]. The simple and rapid capillary zone electrophoresis method was tested for the analysis of alkaloids in the areca nut extract (water sonication) and saliva from areca nut chewers [24].

The objective of the present study was to determine the volatile compounds of the areca nut by using two different extraction methods (headspace solid-phase microextraction (HS-SPME) and simultaneous hydrodistillation–extraction (SHDE)). SHDE products were tested for antimicrobial activity using the agar disk diffusion method.

## 2. Results and Discussion

### 2.1. HS-SPME Analysis of Areca Nuts

In the first part of this study, the HS-SPME method was optimized. To investigate the conditions of SPME extraction of volatile compounds, the main effects of extraction temperature and extraction time (these variables greatly influence the vapor pressure and equilibrium of the aroma compounds in the headspace of the sample) and their interactions were studied by using a central composite design (CCD) based on the response surface methodology (RSM), including four replicates at the central point.

The extraction time and extraction temperature were studied in the range of 10–90 min and 40–120 °C, respectively. For each extraction run, 250 mg of crushed areca nuts was placed in a 20 mL headspace vial sealed with a screw cap equipped with a Teflon septum. The entire matrix of the experimental design with experimental values is summarized in Table 1. Optimal extraction conditions were evaluated in STATISTICA software, version 12 (StatSoft CR, Prague, Czech Republic) based on the number of peaks in the chromatogram (NoP) as a response variable.

The results of the multiple regression analysis are presented in Table 2, and the response surface model obtained for NoP is depicted in Figure 1. The model shows the effects of the extraction temperature and extraction time on the number of peaks detected in the chromatograms. The relationship between the independent variables and their responses depicted in Figure 1 is characterized by the following equation:$$\mathrm{NoP}\;=\;-\;55.22135\;+\;2.15T_{ext}\;-\;0.00961T_{ext}^{2}\;+\;0.85885t_{ext}\;-\;0.00617t_{ext}^{2}$$
where *NoP* is the number of peaks in the chromatogram, *T_ext_* is the extraction temperature (°C), and *t_ext_* is the extraction time (min).

The adjusted root square value (R^2^ = 0.994) shows the high reliability of the model. The data in Table 1 demonstrate an extremely close agreement between the experimental and predicted values of the response variables.

To find which tested factors have effect results of extractions, analysis of variance (ANOVA) of the experimental data was performed (see Table 2). The significance of regression coefficients was tested for the *F*-value and the *p*-value. According to the determined values, both extraction parameters (time and temperature) can be considered significant in linear and quadratic terms. The interaction between the extraction time and extraction temperature is considered to be insignificant.

Optimal HS-SPME conditions were evaluated, and the whole extraction process was as follows: Initially, the sample was incubated for 10 min at the extraction temperature, then the fiber was exposed to the sample headspace under the following conditions: extraction time of 70 min, extraction temperature of 112 °C.

### 2.2. Qualitative and Semiquantitative Analysis of Areca Nuts

Analyses were performed using HS-SPME and SHDE. In total, 98 volatile compounds were identified in the areca nut (8 alcohols, 2 alkanes, 25 carbonyl compounds, 10 esters, 13 fatty acids, 33 terpenes and terpenoids, and 7 other compounds, see Table 3). A total of 41 compounds were identified by HS-SPME/GC-MS and 76 compounds by SHDE/GC-MS. A Venn diagram (Figure 2, created using a web application by Bioinformatics and Systems Biology of Gent, URL: http://bioinformatics.psb.ugent.Be/webtools/Venn/, accessed on 9 July 2021) clearly shows how many compounds are common to each sample treatment method and which compounds are unique. The diagram shows that 22, 47, 2, and 2 unique compounds were identified in the samples analyzed S1 (HS-SPME), S2 (SHDE), S3 (hydrolate), and S4 (distillation residue), respectively. On the contrary, only one compound was found in the four analyzed fractions, namely dodecanoic acid. The most common compounds were observed in samples S1 and S2 (15, of which nine were represented only in samples S1 and S2), and in all cases, they belong to the group of oxidized hydrocarbons.

The highest percentage of the amount in all extracts analyzed was formed by the group of fatty acids (up to 96.5% in the distillation residue). A qualitative analysis of crushed areca nuts was performed under optimized conditions using HS-SPME/GC-MS. A total of 41 compounds were identified (Table 3). There are different groups of identified compounds, and most of them belong to terpenes and higher fatty acids. Arecolin, a typical compound in areca nuts [25,26], has also been identified.

In total, 77 compounds were identified by GC-MS analysis in three fractions (SHDE extract, distillation residue, and hydrolate). The list of identified compounds is summarized in Table 3. Based on the evaluation of the relative content of individual substances, it can be said that fatty acids are the dominant volatile compounds in areca nuts. The biggest difference among the three fractions was observed in the content of terpenes, the SHDE extract was rich in terpenes more than the other two fractions, and HS-SPME extract. Terpenes (namely monoterpenes, sesquiterpenes, and terpenoids) made up more than half of the compounds identified in the SHDE extract. The highest relative amount of the SHDE extract had terpinen−4-ol. This compound belongs to the group of monoterpenoids, which is known to be a potent bactericidal agent [27]. Camphor had the second highest relative amount in the SHDE extract, and this terpenoid has a very strong aroma. Camphor exhibits many biological properties; one of them is the antimicrobial effect [28].

The main differences among the extracts are in the profiles of individual groups of compounds. Figure 3 shows the comparison of the chemical composition of areca nut extracts using bubbles whose sizes correspond to the representation of the individual compounds (the percentage peak area method uses the peak area of the target component as a proportion of the total area of all detected peaks). In terms of the number of compounds identified, the SHDE extract was the richest, as can be seen in Figure 3. In this extract, a broad spectrum of different compounds was identified, most of them terpenes and terpenoids (32 out of 72 identified compounds) with RI ranging from 925 to 1547. Fatty acids have the highest relative amount in all four extracts, concretely 65.51 rel.% in HS-SPME extraction, 85.22 rel.% in SHDE extract, 96.55 rel.% in distillation residue, and 48.53 rel.% in hydrolate. In the case of identified acids, fatty acids with RI were observed ranging from 982 to 2135 in the 4 extracts. Dodecanoic acid was identified in all four extracts. Compounds with RI < 900 were not observed in any extract chromatograms due to solvent cuts during chromatographic analysis. The fewest compounds were identified in the distillation residue. Only eight compounds were detected, and fatty acids (dodecanoic acid and tetradecanoic acid) had the highest relative amounts. In the case of hydrolate, it can be seen in Figure 3 a group of compounds with RI > 900 and <1300 (except for two fatty acids with higher RI), and only very volatile compounds were confirmed. In contrast, there is a wide range of compounds with different RIs in HS-SPME extract.

In the distillation residue is identified arecoline, the most famous alkaloid in areca nuts. There is one similar study [19] from 1998 available, in which the authors separated volatile alkaloids using GC-MS. In our study, only arecoline from the alkaloids group was identified, which is consistent with earlier studies [16,19,20]. Our study is the first to deal with the qualitative analysis of volatile compounds in areca nuts using SHDE-GC-MS. From the total number of compounds assigned, it can be concluded that SHDE was more effective than HS-SPME. The SHDE method seems to be a good tool for qualitative characterization of the volatile profiles of areca nuts. The SHDE method has a great advantage because more extracts (the real one) can be obtained in one extraction procedure and can provide a comprehensive overview of the matrix.

### 2.3. Antimicrobial Effect of Distillation Fractions of Areca Nut

The results of antimicrobial activity of the SHDE extract, hydrolate, and distillation residue of areca nuts and antimicrobials as positive controls are presented in Table 4. According to our findings, the SHDE extract obtained from areca nuts shows an antimicrobial effect against each strain included in our study. In general, the antimicrobial effect of the SHDE extract was greater in the case of Gram-positive bacteria, especially in the case of streptococci (inhibition zone of 27.5–40.0 mm), compared to Gram-negative strains or yeast. The most significant antimicrobial effect was observed against *Streptococcus canis* NPK09 with an inhibition zone of 40.0 ± 3.0 mm. This result can be considered as a very significant antimicrobial activity of the sample that even significantly exceeds the antimicrobial effectiveness of the antibiotics that were included in the study for comparison. There is also shown a considerable antimicrobial effect against strains *Streptococcus agalactiae* CCM 6187 and *Streptococcus pyogenes* NPK01 with inhibition zones 28.3 ± 1.5 and 27.5 ± 1.5 mm. These streptococci are very often isolated from the oral cavity of humans and animals [29,30]. Antimicrobial activity against these microorganisms could reduce the risk of dental caries and other problems related to oral hygiene. The lowest antimicrobial effect of the SHDE extract from areca nuts was observed against *Escherichia coli* CCM 3954 (inhibition zone 10.3 ± 1.1 mm). At the same time, the solvent (*n*-hexane) was tested to determine the possible antimicrobial activity. It was found that *n*-hexane has no inhibitory effect on the microorganisms tested in a range of experiment settings.

Several previous studies have described the antimicrobial effect of areca nuts; however, the studies differ in sample preparation by different extraction procedures. For example, the effect of the areca nut obtained by simple extraction with *n*-hexane, ethanol, or water has been described [9,10,31].

The results of our study show a relatively significant proportion of different terpenes in the SHDE extract of the areca nut. Many of these compounds have previously been confirmed as antimicrobial substances, as described in a recent large-scale study of many terpenes [32]. The strong antimicrobial effect of linalool and α-terpineol was also confirmed in another study [33] and even in the case of terpineol, linalool, and carvacrol vapors [34]. The SHDE extract prepared by us contains a confirmed content of linalool, α-terpineol, and terpinen-4-ol in a relatively significant amount and is one of the most represented terpenes of all that have been detected (see Table 3). Furthermore, the antimicrobial effect of eugenol, α-pinene, and β-pinene was confirmed in many studies [35,36,37]. However, the antimicrobial effects of complex samples could be caused by the synergistic effect of many chemical substances, which could mean influencing the overall antimicrobial activity. The SHDE extract also showed a relatively rich proportion of various fatty acids, of which the most significant content was found in the case of tridecanoic acid and pentadecanoic acid (see Table 3). Previous studies have confirmed the antimicrobial effects of these fatty acids against various microorganisms [38,39,40]. It can also be assumed that the overall antimicrobial effect of the areca nut extract is significantly affected by the fatty acid content.

Testing the antimicrobial effect of the hydrolate and distillation residue obtained during the extraction of areca nuts did not reveal its antimicrobial activity. The unproven antimicrobial potential of hydrolate and distillation residue can be explained mainly by the low proportion of individual components and their concentrations in the hydrolate sample compared to the SHDE extract. Thermal degradation of some compounds can also be considered for the distillation residue. Aqueous and methanolic extracts were tested against *Staphylococcus aureus* and *Escherichia coli* in an earlier study [39]. However, the aqueous extract did not inhibit any of the tested bacteria, which is in line with our results.

## 3. Materials and Methods

### 3.1. Chemicals and Materials

The standard solution of *n*-alkane mixture (C8–C20) dissolved in *n*-hexane (concentration: 40 mg·L^−1^) was purchased from Sigma-Aldrich (Prague, Czech Republic). Helium 5.0 was purchased from Linde Gas a.s. (Prague, Czech Republic). Sterile paper discs with a diameter of 6 mm (Oxoid, Basingstoke, UK) were used for antimicrobial testing, and cultures were grown on Mueller–Hinton agar (Oxoid, Basingstoke, UK). The Separon SGX C18 solid-phase extraction (SPE) cartridge (60 μm, 0.5 g) was purchased from Tessek s.r.o. (Prague, Czech Republic). SPME devices for automatic sampling and fiber with a cover of 2 cm and 50/30 μm divinylbenzene/carboxen/polydimethylsiloxane (DVB/CAR/PDMS) were supplied by Supelco (Bellefonte, PA, USA).

### 3.2. Plant Material

A 500 g amount of samples of areca nuts was purchased from the company Herbal-store s.r.o. (Jablonec nad Nisou, Czech Republic). To obtain the ground areca nut sample, the nuts were ground in a knife mill (Retsch, Haan, Germany).

### 3.3. Instrumentation GC-MS and GC-FID and Analysis of Volatile Compounds

Gas chromatography coupled with the QP 2010 Plus mass spectrometer (MS) and flame ionization detector (FID) was performed using a gas chromatograph GC2010 (Shimadzu, Kyoto, Japan) fitted with a CombiPAL autosampler (CTC Analytics, AG, Zwingen, Switzerland). Conditions of the measurements are summarized below, and they were set the same for both GC-MS (used for identification of volatile compounds) and GC-FID (used for semi-quantification of volatile compounds) analysis, respectively.

GC conditions: A SLB-5MS capillary column (length: 30 m, inner diameter: 0.25 mm, film thickness: 0.25 μm, Supelco, Bellefonte, PA, USA) was used for GC separation. Helium 5.0 was used as the carrier gas at a constant linear velocity of 30 cm·s^−1^. The split ratio was set to 1:5. The injector temperature was maintained at 200 °C. The temperature gradient was programmed as follows: the initial temperature was 40 °C (3 min) and then increased at 5 °C min^−1^ to 250 °C with a total analysis time of 60 min.

FID conditions: The detector temperature was set to 270 °C.

MS conditions: The interface temperature and the ion source temperature were both maintained at 200 °C. The mass spectrometer was operated in electron ionization mode (70 eV), and the detection of ions was performed in full-scan mode in a range of 35–500 *m*/*z*. The identification of the compounds was based on a comparison of the obtained mass spectra with reference mass spectra from the NIST 14 (NIST, Gaithersburg, MD, USA) and FFNSC 2 (Shimadzu, Kyoto, Japan) libraries and by comparing their retention indices of the compounds. Retention indices were calculated according to the equation of van den Dool and Kratz [41] for a homologous series of *n*-alkanes injected under the same GC conditions. These retention indices were verified by comparing the retention indices with published data through NIST WebBook Chemistry [42].

### 3.4. Headspace Solid-Phase Microextraction Technique

Headspace SPME sampling was performed using CTC combiPAL, which is designed for automated sampling. The SPME fiber was conditioned according to the manufacturer’s instructions.

The HS-SPME method was performed as follows: 250 mg of ground areca nut sample was weighed into a 20 mL headspace vial, and the vial was closed with a magnetic cap with a Teflon septum. The sample was incubated for 10 min at 112 °C, and sorption of volatile compounds was carried out for 70 min at 112 °C on a 50/30 μm divinylbenzene/carboxen/polydimethylsiloxane (DVB/CAR/PDMS) fiber. Desorption was carried out in the injection port of a gas chromatograph at 200 °C for 30 s. Any possible carry-over effect between individual injections was eliminated by cleaning of the fiber in the SPME fiber conditioning module at 250 °C for 10 min.

### 3.5. Simultaneous Hydrodistillation–Extraction (SHDE)

SHDE was performed using an apparatus of Clevenger type (Kavalierglass a.s., Prague, Czech Republic). A 40 g amount of crushed areca nuts was distilled with 500 mL of distilled water for 5 h. Volatilized compounds and water vapor were condensed through a cooling system, collected, and simultaneously extracted using 1 mL of *n*-hexane in a separator tube (SHDE extract = *n*-hexane extract). The SHDE procedure was performed three times. The byproducts of SHDE (hydrolate and distillation residues in the distillation flask) were also collected and analyzed.

### 3.6. Treatment of SHDE Byproducts before GC Analysis

Briefly, the hydrolate and the distillation residue were treated as follows: 10 mL of liquid sample was loaded on SPE cartridge, and the sorbed compounds were eluted with 2 mL of *n*-hexane. The procedure was replicated three times for each byproduct, and the appropriate eluents were mixed.

### 3.7. Antimicrobial Testing

The antimicrobial susceptibility of several strains (*Bacillus subtilis* CCM 2215, *Enterococcus faecalis* CCM 4224, *Escherichia coli* CCM 3954, *Pseudomonas aeruginosa* CCM 3955, *Staphylococcus aureus* CCM 4223, *Streptococcus agalactiae* CCM 6187, *Streptococcus canis* NPK09, *Streptococcus pyogenes* NPK01 and *Candida albicans* CCM 8215) to SHDE extract, hydrolate, and the distillation residue of the areca nut was tested by a previously described disk diffusion method [43]. The strains were obtained from the Czech Collection of Microorganisms, Brno, Czech Republic (CCM), or provided by the Pardubice Hospital (NPK) as clinical isolates.

Briefly, strains were grown on Mueller–Hinton agar (Oxoid, Basingstoke, UK) or Sabouraud dextrose agar (HiMedia, India) in the case of *Candida albicans*. Cells were harvested and suspended in physiological saline to approximately a cell density of 1.5 × 10^8^ CFU·mL^−1^ (0.5 on the McFarland scale). The microbial cell suspensions were spread onto Mueller–Hinton or Sabouraud dextrose agar using sterile cotton swabs. Sterile paper disks (6 mm in diameter) were placed on the inoculated agar medium and impregnated with 8 μL of sample. Plates were incubated for 24–48 h at 30–37 °C (depending on strain) under aerobic conditions. Following incubation, inhibition zones were evaluated using a BACMED 6iG2 automated reader and analyzer (Aspiag, Czech Republic). Simultaneously, antimicrobial disks were purchased from Oxoid Ltd. (Basingstoke, UK) were used as a positive control (ampicillin: 10 µg, ciprofloxacin: 5 µg, clindamycin: 2 µg, fluconazole: 25 µg, and tetracycline: 30 µg). As a negative control for SHDE extract, testing with pure *n*-hexane (solvent) was performed. Experiments were carried out repeatedly (*n* = 4), and the results were expressed as the mean of inhibition zones (in millimeters, including 6 mm of disk diameter) with a calculated standard deviation.

## 4. Conclusions

The areca nut is the fourth most used drug in the world, so it is very beneficial to know what compounds are found inside. The compounds of the areca nut were extracted in two different ways, concretely using HS-SPME and SHDE. A total of three samples (distillation residue, hydrolate, and SHDE extract) were obtained by SHDE. These extracts were separated and identified by GC-MS. The obtained spectra were compared with a library of reference MS spectra, and to confirm the correctness of the identification, the obtained retention indices were compared with the reference retention indices. During all our experiments, we identified in total 98 volatile compounds. The main groups of identified substances were alkanes, alcohols, aldehydes, esters, fatty acids, ketones, and terpenes. Furthermore, arecoline, the main alkaloid of areca nuts, was found. The extracts obtained using SHDE were tested for their antibacterial activity. Only the SHDE extract was found to have antibacterial effects; there were no inhibition zones for samples of the distillation residue and hydrolate. The SHDE extract had the greatest antibacterial effect against the strain *Streptococcus canis*. In conclusion, we can say that the areca nuts are an outstanding natural matrix due to the number of compounds. Our study showed great potential for using areca nuts as antimicrobial agents, but it must be studied thoroughly. Furthermore, it is necessary to study the possible cytotoxicity and carcinogenicity.

## Figures and Tables

**Figure 1 molecules-26-07422-f001:**
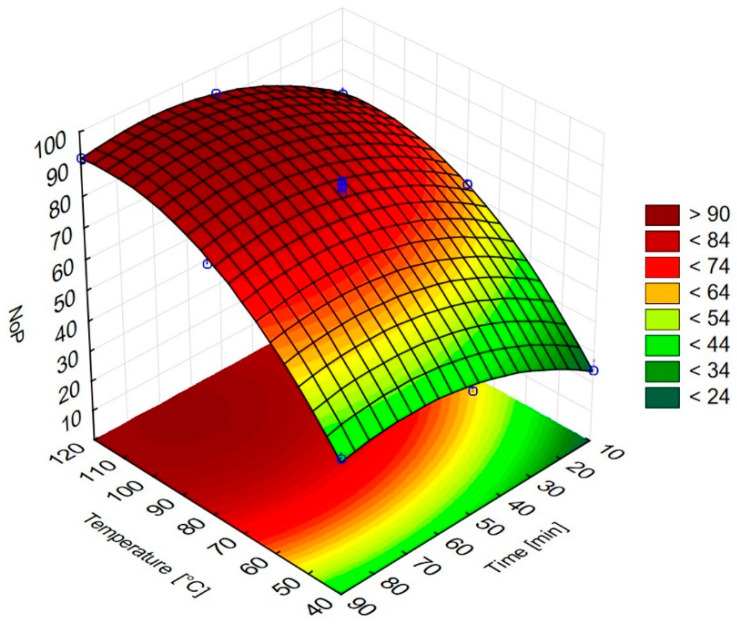
Dependence of the number of peaks on the extraction conditions.

**Figure 2 molecules-26-07422-f002:**
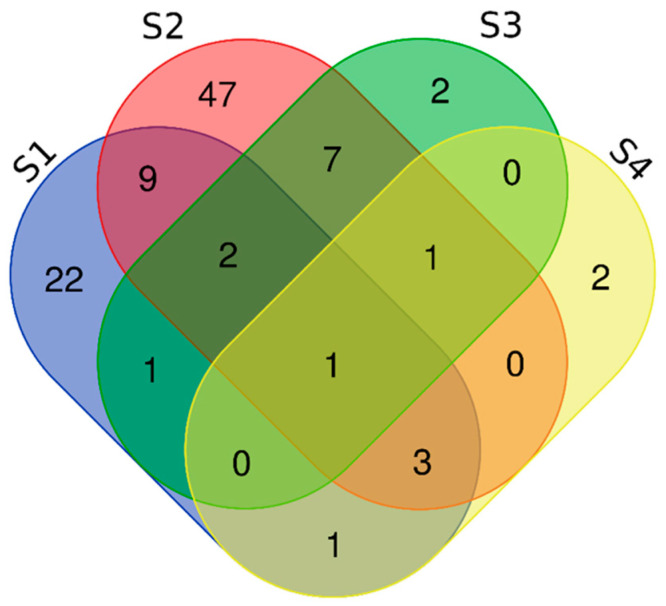
Venn diagram: comparison of the identified compounds in the four fractions obtained using different treatment methods for the areca nut sample.

**Figure 3 molecules-26-07422-f003:**
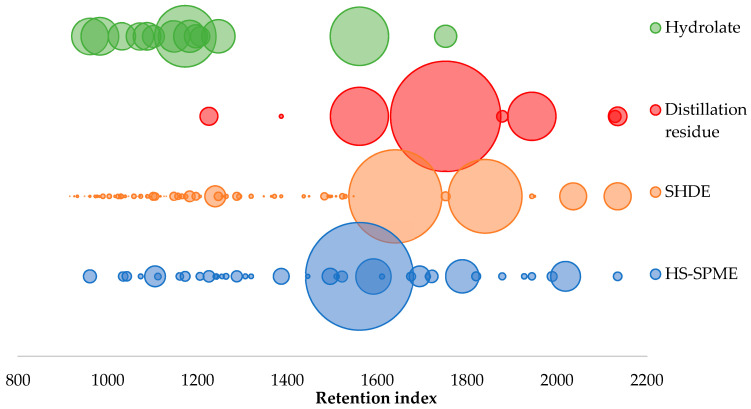
Comparison of the chemical composition of areca nut extracts using bubbles whose sizes correspond with the representation of the individual compounds (the percentage peak area method uses the area of the target component peak as a proportion of the total area of all detected peaks).

**Table 1 molecules-26-07422-t001:** Experimental conditions used for the central composite design analysis and the corresponding observed and predicted responses for HS-SPME *(T_ext_* = extraction temperature; *t_ext_* = extraction time; (*C*) = central point; *NoP* = number of peaks in the chromatogram).

Run	Coded Factors		Decoded Factors		Response	
	*T_ext_*	*t_ext_*	*T_ext_* (°C)	*t_ext_* (min)	NoP	
					Observed	Predicted
1	−1	−1	40	10	24	23
2	−1	1	40	90	44	43
3	−1	0	40	50	41	43
4	0	−1	80	10	63	63
5	0	1	80	90	81	83
6 (C)	0	0	80	50	84	83
7 (C)	0	0	80	50	85	83
8 (C)	0	0	80	50	83	83
9 (C)	0	0	80	50	81	83
10	1	−1	120	10	72	72
11	1	1	120	90	92	92
12	1	0	120	50	92	92

**Table 2 molecules-26-07422-t002:** Coefficients of the central composite design for NoP fitted with the second-order polynomial model and analysis of variance (ANOVA) for the experimental results.

Parameter	Regression Coefficient	Standard Error	Degree of Freedom	Sum of Squares	*F*-Value	*p*-Value
Lack-of-fit			3	9.375	1.071	0.478052
Pure error			3	8.750		
b0	−55.2214	4.407486				
b1	2.1500	0.109332	1	3601.500	1234.800	0.000051
b11	−0.0096	0.000654	1	630.375	216.129	0.000683
b2	0.8589	0.079995	1	560.667	192.229	0.000812
b22	−0.0062	0.000654	1	260.042	89.157	0.002518
b12	0.0000	0.000534	1	0.000	0.000	1.000000

**Table 3 molecules-26-07422-t003:** Identified compounds in analyzed samples obtained using HS-SPME and SHDE of areca nuts.

GROUP OF COMPOUNDS Volatile Compounds	Retention Index	CAS Number	Relative % of Peak Area			
			HS-SPME Extraction	SHDE Extract	Distillation Residue	Hydrolate
**ALKANES**						
Hexadecane	1597	544-76-3	0.16	-	-	-
Nonadecane	1897	629-92-5	0.31	-	-	-
**∑** **ALKANES**			**0.47 (2)**	**-**	**-**	**-**
**ALCOHOLS**						
1-Heptanol	972	111-70-6	-	0.07	-	-
1-Octen-3-ol	981	3391-86-4	-	0.05	-	-
3-Octanol	998	589-98-0	-	0.01	-	-
Benzenemethanol	1034	100-51-6	0.56	-	-	-
1-Octanol	1073	111-87-5	0.18	0.13	-	-
Benzeneethanol	1111	60-12-8	0.31	-	-	-
1-Methyl-4-(1-methylethyl)-cis-2-cyclohexen-1-ol	1125	29803-82-5	-	0.01	-	-
1-Hexadecanol	1878	36653-82-4	0.34	-	-	-
**∑ ALCOHOLS**			**1.39 (4)**	**0.27 (5)**	**-**	**-**
**ALDEHYDES**						
*trans*-2-Heptenal	957	18829-55-5	-	0.01	-	-
Benzaldehyde	960	100-52-7	1.02	0.05	-	7.23
Octanal	1003	124-13-0	-	0.16	-	-
Phenylacetaldehyde	1042	122-78-1	0.56	0.01	-	-
Nonanal	1105	124-19-6	2.45	0.38	-	-
trans-2-Nonenal	1160	18829-56-6	0.37	0.09	-	-
Decanal	1205	112-31-2	0.40	0.07	-	2.16
*p*-Isopropylbenzaldehyde	1241	122-03-2	0.23	-	-	-
Anisaldehyde isomer	1254	-	0.17	-	-	-
*trans*-2-Decenal	1263	3913-81-3	0.24	0.15	-	-
Undecanal	1306	112-44-7	0.19	-	-	-
*trans*-2, *trans*-4-Decadienal	1319	25152-84-5	0.17	0.14	-	-
2-Undecenal	1364	2463-77-6	-	0.04	-	-
2-Butyl-2-octenal	1371	13,019-16-4	-	0.13	-	-
Tridecanal	1509	10486-19-8	0.22	-	-	-
Tetradecanal (Myristaldehyde)	1610	124-25-4	0.20	-	-	-
1-Pentadecanal	1712	2765-11-9	0.22	-	-	-
**∑ALDEHYDES**			**6.44 (13)**	**1.23 (11)**	**-**	**9.39 (2)**
**ESTERS**						
Amyl acetate	915	628-63-7	-	0.01	-	-
Dodecanoic acid, methyl ester	1521	111-82-0	0.75	-	-	-
Dodecanoic acid, ethyl ester	1591	106-33-2	6.80	-	-	-
Tetradecanoic acid, methyl ester	1721	124-10-7	0.96	-	-	-
Tetradecanoic acid, ethyl ester	1789	124-06-1	6.04	-	-	-
Tetradecanoic acid, 1-methylethyl ester	1820	110-27-0	0.51	-	-	-
Amyl laurate	1878	5350-03-8	-	-	0.82	-
Hexadecanoic acid, methyl ester	1927	112-39-0	0.22	-	-	-
Hexadecanoic acid, ethyl ester	1989	628-97-7	0.61	-	-	-
Hexadecanoic acid, 1-methylethyl ester	2019	142-91-6	4.88	-	-	-
**∑ ESTERS**			**20.77 (8)**	**0.03 (2)**	**0.82 (1)**	**-**
**FATTY ACIDS**						
Hexanoic acid	982	142-62-1	-	-	-	7.65
Heptanoic acid	1093	111-14-8	-	0.03	-	-
Octanoic acid	1172	124-07-2	0.61	-	-	20.0
Nonanoic acid	1239	112-05-0	0.19	2.48	-	-
Decanoic acid	1386	334-48-5	1.47	0.08	0.13	-
Undecanoic acid	1436	112-37-8	-	0.09	-	-
Dodecanoic acid	1560	143-07-7	60.40	45.23	17.88	18.1
Tetradecanoic acid	1752	544-63-8	-	0.51	63.24	2.78
Pentadecanoic acid	1840	1002-84-2	-	28.56	-	-
Hexadecanoic acid	1944	57-10-3	0.36	0.16	12.43	-
Heptadecanoic acid	2036	506-12-7	-	4.01	-	-
*cis*-9, *cis*-12-Octadecadienoic acid	2129	60-33-3	-	-	0.88	-
*trans*-9-Octadecenoic acid	2135	112-80-1	0.48	4.07	1.99	-
**∑ FATTY ACIDS**			**63.51 (6)**	**85.22 (10)**	**96.55 (6)**	**48.53 (4)**
**KETONES**						
Oct-3-en-2-one	1038	1669-44-9	-	0.07	-	-
Acetophenone	1065	98-86-2	-	0.01	-	-
2-Undecanone	1292	112-12-9	-	0.20	-	-
Nerylacetone	1445	3879-26-3	0.14	-	-	-
2-Tridecanone	1495	593-08-8	1.52	0.07	-	-
3-ethenyl-3-methyl-6-(1-methylethyl)-2-(1-methylethylidene)- cyclohexanone	1528	21698-46-4	-	0.14	-	-
*γ*-Dodecalactone	1675	2305-05-7	0.50	-	-	-
2-Pentadecanone	1694	2345-28-0	2.43	-	-	-
**∑ KETONES**			**4.59 (4)**	**0.49 (5)**	**-**	**-**
**TERPENES and TERPENOIDS**						
*α*-Thujene	925	2867-05-2	-	0.02	-	-
*α*-Pinene	932	80-56-8	-	0.06	-	-
Camphene	948	79-92-5	-	0.01	-	-
*β*-Pinene	976	127-91-3	-	0.06	-	-
*α*-Terpinen	1016	99-86-5	-	0.06	-	-
*p*-Cymene	1023	99-87-6	-	0.16	-	-
Limonene	1029	138-86-3	-	0.17	-	-
Eucalyptol	1031	470-82-6	-	0.11	-	4.11
*β*-Ocimene	1046	13877-91-3	-	0.01	-	-
*cis*-Linalool oxide	1071	5989-33-3	-	0.02	-	4.10
Terpinolene	1085	586-62-9	-	0.02	-	-
*trans*-Linalool oxide	1087	34995-77-2	-	-	-	4.19
Fenchone	1088	1195-79-5	-	0.11	-	-
Linalool	1101	78-70-6	-	0.41	-	2.73
Thujone	1117	546-80-5	-	0.03	-	-
Camphor	1147	76-22 -2	-	0.43	-	5.38
Isomenthone	1156	491-07-6	-	0.28	-	-
D-isomenthone	1165	1196-31-2	-	0.18	-	-
Menthol	1171	89-78-1	-	0.02	-	-
L-Borneol	1173	464-45-9	-	0.15	-	-
Terpinen-4-ol	1182	562-74-3	-	0.71	-	5.60
*α*-Terpineol	1196	98-55-5	-	0.40	-	3.01
D-Carvone	1246	2244-16-8	-	0.46	-	6.06
Geraniol	1253	106-24-1	-	0.02	-	-
Piperitone	1255	89-81-6	-	0.04	-	-
*α*-Terpinyl acetate	1347	80-26-2	-	0.02	-	-
Geranyl acetone	1448	3796-70-1	-	0.03	-	-
*α*-Curcumene	1482	644-30-4	-	0.35	-	-
*β*-Selinene	1491	17066-67-0	-	0.08	-	-
*α*-Selinene	1498	473-13-2	-	0.05	-	-
*β*-Bisabolene	1508	495-61-4	-	0.03	-	-
Myristicin	1522	607-91-0	-	0.22	-	-
Elemicin	1547	487-11-6	-	0.02	-	-
**∑ TERPENES and TERPENOIDS**			**-**	**4.74 (32)**	**-**	**35.18 (8)**
**OTHER COMPOUNDS**						
2,6,6-trimethyl-2-ethenyltetrahydropyran	969	7392-19-0	-	0.01	-	-
2-pentylfuran	989	3777-69-3	-	0.16	-	-
4-ethenyl-1,5,5-trimethylcyclopentene	1058	1727-69-1	-	0.16	-	-
4-Acetyl-1-methylcyclohexene	1130	6090-09-1	-	0.01	-	-
Arecoline	1225	63-75-2	0.80	-	1.85	-
Anethole	1287	4180-23-8	0.76	0.42	-	-
6-Nonyltetrahydro-2H-pyran-2-one	1950	2721-22-4	-	0.05	-	-
**∑ OTHER COMPOUNDS**			**1.56 (2)**	**0.83 (7)**	**1.85 (1)**	

(∑ in table is expressed always for each group of compounds; first number is the sum of % rel. and the second number in () is number of compounds).

**Table 4 molecules-26-07422-t004:** Antimicrobial activity of SHDE extract, hydrolate, and distillation residue of areca nut sample and antimicrobials as positive control—mean inhibition zones in mm (including disc 6 mm in diameter) ± standard deviation, *n* = 4.

	SHDE Extract	Hydrolate	Distillation Residue	AMP 10 µg	CIP 5 µg	DA 2 µg	TE 30 µg	FCA 25 µg
*Bacillus subtilis* CCM 2215	12.3 ± 1.5	6.0 ± 0	6.0 ± 0	21.5 ± 1.5	35.0 ± 0	25.5 ± 0.5	22.5 ± 0.5	n.t.
*Enterococcus faecalis* CCM 4224	16.0 ± 0.7	6.0 ± 0	6.0 ± 0	12.5 ± 0.5	19.0 ± 0.0	8.0 ± 0.0	23.0 ± 1.0	n.t.
*Escherichia coli* CCM 3954	10.3 ± 1.1	6.0 ± 0	6.0 ± 0	6.0 ± 0.0	21.5 ± 1.5	6.0 ± 0.0	13.5 ± 7.5	n.t.
*Pseudomonas aeruginosa* CCM 3955	11.3 ± 0.8	6.0 ± 0	6.0 ± 0	6.0 ± 0.0	36.0 ± 1.0	6.0 ± 0.0	14.0 ± 1.0	n.t.
*Staphylococcus aureus* CCM 4223	10.8 ± 0.8	6.0 ± 0	6.0 ± 0	28.5 ± 1.5	29.0 ± 1.0	30.0 ± 1.0	16.5 ± 0.5	n.t.
*Streptococcus agalactiae* CCM 6187	28.3 ± 1.5	6.0 ± 0	6.0 ± 0	17.5 ± 0.5	20.5 ± 0.5	22.5 ± 0.5	14.5 ± 0.5	n.t.
*Streptococcus canis* NPK09	40.0 ± 3.0	6.0 ± 0	6.0 ± 0	21.5 ± 1.5	21.5 ± 2.5	22.5 ± 0.5	11.0 ± 1.0	n.t.
*Streptococcus pyogenes* NPK01	27.5 ± 1.5	6.0 ± 0	6.0 ± 0	31.5 ± 1.5	28.5 ± 1.5	34.0 ± 0.0	37.0 ± 0.0	n.t.
*Candida albicans* CCM 8215	15.0 ± 1.7	6.0 ± 0	6.0 ± 0	n.t.	n.t.	n.t.	n.t.	11.5 ± 0.5

AMP—ampicillin, CIP—ciprofloxacin, DA—clindamycin, TE—tetracycline, FCA—fluconazole, n.t.—not tested.

## Data Availability

Not applicable.

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
