# Peer review of "Volatiles Composition and Antimicrobial Activities of Areca Nut Extracts Obtained by Simultaneous Distillation–Extraction and Headspace Solid-Phase Microextraction"

_molecules, 2021, doi:10.3390/molecules26247422_

Round 1

Reviewer 1 Report

The study investigated the areca nut for its volatile compounds. Traditionally, areca nut has been used as an aphrodisiac, appetite suppressant, digestive aid and diuretic; and as a treatment for asthma, cough, dermatitis, glaucoma, impotence, intestinal worms and toothache, etc. I consider it important that the field better understand the properties of volatile compounds, as it represents a scientific advance in a species that many people are used to using. The aim of the present study was to determine the volatile compounds of the areca nut using two different extraction methods. SHDE (simultaneous hydrodistillation extraction) products were tested for antimicrobial activity by the agar disk diffusion method against various bacteria and this is important considering that this nut has several pharmacological properties already described, in addition to being widely used by the oriental population. So I am in favor of the publication.

Author Response

We have no comments to Reviewer 1.

Reviewer 2 Report

The study is quite impressive and scientifically elaborated. I advise some minor corrections:

- In the material and methods section, please add the description of all protocols (parameters and conditions of analysis to make the tests more visible for lecturers):

- Please, improve the quality of figures.

- Ensure the uniformity in the units (e.g. mg/mL, µL) throughout the MS.

- Correct some English mistakes before publication.

- Check the references in accordance with the journal style.

Author Response

C - comment: In the material and methods section, please add the description of all protocols (parameters and conditions of analysis to make the tests more visible for lecturers):

A - answer: Chapter 3.4 was rewritten and now should be more visible how the experiments were done.

C: Please, improve the quality of figures.

A: better quality figure 1 was inserted.

C: Ensure the uniformity in the units (e.g. mg/mL, µL) throughout the MS.

A: We checked all MS and changed units to correct form.

C: Correct some English mistakes before publication.

A: English was checked and English mistakes were corrected.

- Check the references in accordance with the journal style.

References were checked and prepared according with the journal style.

Reviewer 3 Report

The manuscript entitled "Qualitative analysis of volatile substances released from areca nuts and testing of antimicrobial properties of the product of simultaneous hydrodisillation extraction" presents volatile organic compounds determination in Areca nut. The manuscript has a significant flaws in experimental procedures and manuscript preparation.

The abstract in not prepare in the well-manner. All information about results, methodology and aim is mixed.

I am not an English native speaker, however in my opinion there is a massive problem with the English language, even the title doesn't sound properly.

Moreover, the methodology used is insufficient to be published in Journal with high IF. Were any chemical standards or internal standards applied? The identification solely based on the NIST library and retention indices, especially by the inexperienced user, is not enough for being called "identification". The identification and quantification are a crucial steps to achieve valuable data, that can be further published. 

Any statistical tool is exploited in this study. Due to that there cannot be said is there are significant differences in extraction procedures.

The discussion doesn't based on the up-to-date literature, some of cited articles are from mid 90's.

Therefore, conclusions are overestimated according to these analysis.

I am not capable to recommend this work for further processing, and my decision is REJECT.

Author Response

Comment: The abstract in not prepare in the well-manner. All information about results, methodology and aim is mixed.

Answer: Abstract was completely rewritten.

C: I am not an English native speaker, however in my opinion there is a massive problem with the English language, even the title doesn't sound properly.

A: English was checked and English mistakes were corrected. Title of MS was rewritten.

C: Moreover, the methodology used is insufficient to be published in Journal with high IF. Were any chemical standards or internal standards applied? The identification solely based on the NIST library and retention indices, especially by the inexperienced user, is not enough for being called "identification". The identification and quantification are a crucial steps to achieve valuable data, that can be further published.

A: In publications of this type, these methods are the usual way of identifying organic compounds. We agree that some substances could be verified using standards, but for most substances this is not the case due to the unavailability of standards. In our case, the FFNSC2 library was used for identification (among others), which was created from data measured on the same chromatographic column that was also used in our research. Combined with the retention index method and the experience of the researcher, this is a powerful combination for the identification of organic compounds.

C: Any statistical tool is exploited in this study. Due to that there cannot be said is there are significant differences in extraction procedures.

A: The results of the statistical evaluation were added to optimize the SPME. The main purpose of this research was to identified as much as possible compounds in area nuts by using of chosen techniques. A Venn diagram was created to compare extraction techniques in terms of identified compounds.

C: The discussion doesn't based on the up-to-date literature, some of cited articles are from mid 90's.

A: There were used available literature sources. This matrix was not examined from the point of view of qualitative analysis, therefore older literature sources are also used.

Reviewer 4 Report

The paper presents some interesting results, but some improvements are needed.

Line 76-78. What is the DOE used. After is understood that is CCD, but better to begin to explain that.

Line 86 Begin with the equation from the analysis of CCD but only after presenting the table. Must be reformulated the order of presentation.

Figure 1 must be improved to see better the axis XX oy YY.

All the factors have the same significance. Maybe some ANOVA parameters can be clearer?

All this part needs to be restructured.

Line 153 to 154. The comparison of the techniques must be done carefully. The SHDE must present differences to the HS-SPME so is clear that the first one must present more compounds in the composition but with different % between volatile or fatty acids.

Why in the experimental part of Antimicrobial testing only a disc was used?

IN the references the volume number must be in bold?

In the references is missing many DOI.

Author Response

Comment: Line 76-78. What is the DOE used. After is understood that is CCD, but better to begin to explain that.

Answer: This information was added to MS.

C: Line 86 Begin with the equation from the analysis of CCD but only after presenting the table. Must be reformulated the order of presentation.

A: Appropriate section was rewritten.

C: Figure 1 must be improved to see better the axis XX oy YY.

A: Figure was added in better quality.

C: All the factors have the same significance. Maybe some ANOVA parameters can be clearer?

A: ANOVA parameters were added in new table together with text information.

C: Line 153 to 154. The comparison of the techniques must be done carefully. The SHDE must present differences to the HS-SPME so is clear that the first one must present more compounds in the composition but with different % between volatile or fatty acids. A Venn diagram was created to compare extraction techniques in terms of identified compounds

C: Why in the experimental part of Antimicrobial testing only a disc was used?

A: It is a standard methodology for assessing the antimicrobial potential of substances and comparing the antimicrobial effect of individual samples on a range of selected microorganisms.

C: IN the references the volume number must be in bold?

A: Format of volume number was changed in all references according to the journal recommendations.

C: In the references is missing many DOI.

A: We added DOI numbers to the references (if available)

Round 2

Reviewer 3 Report

The Authors indeed made some improvements, therefore I advise major revision and suggest working still on the manuscript.

In the title: there is "chemical composition" whereas it is volatiles composition.

HS-SPME is not the method to obtain extracts, but the method of extraction.

The abstract doesn't contain the presentation of the results.

I suggest also working on the keywords. This will be helpful to identify the article for readers. 

Did authors used any internal standard or standards of volatiles? Basis of the retention indices is very little. 

Moreover, Authors didn't present any statistical methods for their results.

In table 3, I recommend to put the volatiles in the alphabetical order.

Some of the references lacks of the DOI number.

Reviewer 4 Report

The paper has been improved and revised according to what is suggested.

It can be accepted.

Just check if the references the DOI is in capital or doi.

Author Response

DOI should be capital.